# Sustainability Analysis of the Production of Early Stages of the Atlantic Forest Lambari (*Deuterodon iguape*) in a Public Hatchery at a Rainforest Conservation Area

**Dalton Belmudes [1], Fernanda S. David [1,2], Fernando H. Gonçalves [3]** and **Wagner C. Valenti [1,*]**

1   Aquaculture Center and CNPq, UNESP—São Paulo State University, São Paulo 14884-900, Brazil; dalton.belmudes@gmail.com (D.B.); fselesdavid@hotmail.com (F.S.D.)
2   RINA Brasil Serviços Técnicos LTDA, São Paulo 04080-001, Brazil
3   Virginia Seafood Agricultural and Extension Center, SEAMaR, CAIA and CCS, Virginia Tech—Virginia Polytechnic Institute and State University, Hampton, VA 23669, USA; ocfernando@vt.edu
*   Correspondence: w.valenti@unesp.br

**Abstract:** Protected areas have been used worldwide to conserve natural resources. Nevertheless, economic activities to provide income for communities living within and surrounded by conservation areas remain an issue. This study aimed to assess the sustainability of a *Deuterodon iguape* hatchery, situated within an Atlantic Rainforest Park, to leverage grow-out farming of this small native fish, affording income and food security for local families. We have used a set of indicators of economic, social, and environmental sustainability. The initial investment is about US$ 40,000, which should see a return in ~2 years. The internal rate of return is close to 50%, including the externality costs, which is attractive for both public and private investors. The hatchery generated few direct jobs, but the workforce can be recruited from the community, and the hatchery can enable the establishment of several small grow-out farms, leveraging the development of indirect jobs and self-employment. The system had a low environmental impact, showing a minor release of pollutants, a low risk for biodiversity, and absorption of 18 g of $CO_2$ equivalent per thousand post-larvae produced, contributing to the struggle against climate change. Therefore, the *D. iguape* hatchery demonstrates the potential of combining biodiversity conservation and income generation, meeting the Sustainable Development Goals of Agenda 2030.

**Keywords:** sustainability; hatchery; SDG; lambari; conservation unit; rural farm; *Deuterodon iguape*



## 1. Introduction

In recent decades, protected areas have been used worldwide to preserve and conserve natural resources. In Brazil, different management categories were established by legislation through the National System of Conservation Units [1]. Integral Protection Conservation Units, such as the Sea Mountains State Park, do not allow residents' settlement to any great extent. However, the areas were either already inhabited before the regulatory decree or had later occupation by residents. Conflicts of interest and the lack of links with other sectoral policies compromise the management of these protected areas and caused delays in issuing management plans. The Sea Mountains State Park had its management plan drawn up 30 years after its consolidation as a Conservation Unit. The enforcement of the regulations to mitigate environmental problems impacted many families and small farmers from surrounding buffer areas, who lost their sources of income and livelihood, causing negative socio-economic issues.

In this context, several initiatives to generate income and improve the quality of life of the local communities have been attempted, such as fish production. This business promotes food production, gives nutritional security to the farmer's family, improves the use of the available resources, and generates income [2]. In addition, fish production can

be a predictable and constant source of food, available throughout the year. Globally, aquaculture has been a significant contributor to socio-economic development over the past few years. However, the current challenge is to reconcile social, economic, and environmental development in rural communities [3,4]. The culture of small native fish of low trophic levels can match this purpose [5]. These fish can be farmed in simple facilities, using cheap food, and frequently are eaten whole with the bones, heads, and viscera. Thus, their culture uses a low amount of natural and economic resources to provide nutrients for vulnerable populations [5–7]. In addition, the use of native species reduces the environmental impact of leakage and the spread of pathogens [8,9].

Among the native small fish in Brazil, the Atlantic forest lambari, *Deuterodon iguape*, has great potential for sustainable production, promoting socio-economic development, improving food security, and conserving natural resources [5]. This fish occurs in most bodies of water in the Sea Mountains State Park. It has suitable characteristics for aquaculture, such as easy reproduction, rapid larvae development and juvenile growth, and simple management. The Atlantic forest lambari is sold as baitfish for sport fishing and human consumption around the Sea Mountains State Park [9,10]. Lambari can also be transported to and traded in other regions with the market appeal of the Sea Mountains State Park's territorial product.

The farming of *D. iguape* began in the Sea Mountains State Park, with the capture of fingerlings in local rivers for grow-out in ponds. Nevertheless, the low number of seeds captured, and the irregular seed supply, could not sustain commercial production. This scenario has led to the culture of exotic species in many farms around the Sea Mountains State Park, generating environmental conflicts. Thus, the steady production of post-larvae of *D. iguape* emerged as an alternative to provide this essential input for developing the aquaculture in the region. With well-defined reproduction methods, public or private stakeholders can produce post-larvae to supply local demands for lambari grow-out producers to leverage an economic activity based on a native species that does not compromise the park or the buffer areas.

Sustainable aquaculture embodies economic, social, and environmental aspects and should see profitable production, harmonious interaction with local communities, fair remuneration, and conservation of the environment [4]. It is essential to measure sustainability, to assess the strengths and weaknesses of aquaculture production systems. The economic dimension includes production costs, revenues, project liquidity, externalities, and investment feasibility [4]. The social dimension relates to food security, the equitable distribution of income, the generation of jobs, and self-employment [4]. The environmental dimension relates directly to the ecosystems that provide services and inputs to the activity and receive waste products; thus, the major aspects are maintaining biodiversity, not compromising or depleting natural resources, and not significantly altering the functioning of the surrounding ecosystems [11].

Sustainability can be assessed using indicators to characterize different production process features in a simplified way [11]. Sustainability indicators have analytical effectiveness and provide essential information to assess and indicate the needed improvements in the aquaculture systems. They can also assist policymakers and entrepreneurs in their decisions. Therefore, the objective of this work was to measure the sustainability of the larviculture of *D. iguape* at the Guanhanhã Municipal Aquaculture Farm, using a set of indicators to generate information to supports the sustainable development of aquaculture in the Atlantic forest region. In addition, we tested the hypothesis that profitable and sustainable hatcheries of native fish species can be installed close to protected areas to leverage aquaculture as an economic activity with low environmental impact.

## 2. Materials and Methods

Data were collected in the hatchery of *D. iguape* placed at the Guanhanhã Municipal Aquaculture Farm (24°12′26.12″ S, 47°2′48.24″ W) (Figure 1), located within the Sea Mountains State Park, Itariru nucleus, São Paulo State, Brazil. The analyses of the environmental,

social, and economic sustainability were carried out using the set of indicators developed by Valenti et al. [4]. The mathematical principles and formulae to calculate each indicator are described in that article [4], and, therefore, they are not repeated in this study.

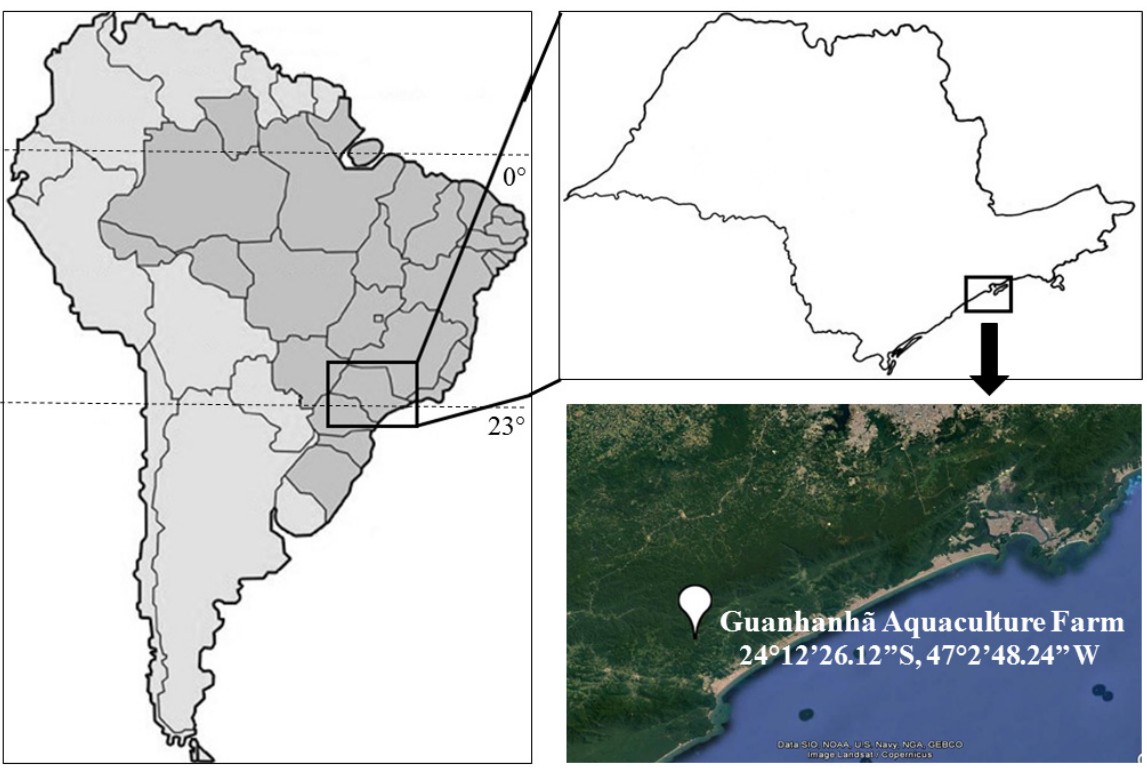

**Figure 1.** Location of the Guanhanhã Municipal Aquaculture Farm (24°12′26.12″ S, 47°2′48.24″ W), located within the Sea Mountains State Park, Itariru nucleus, São Paulo State, Brazil. From Fonseca et al. [5].

### 2.1. Reproduction and Larviculture Procedures

The procedures described below (Figure 2) are regularly performed at the Guanhanhã Municipal Aquaculture Farm and can be easily adapted to any hatchery. The site has two 100 m² earthen ponds to maintain a broodstock of about 2000 fish, and three 1000 L tanks, with four spawning boxes inside each one. Sixty females and 120 males were collected in the advanced maturation stage in the earthen ponds and transferred to the laboratory. Five females were stocked in each of the 12 spawning boxes. The temperature was maintained between 25 °C and 27 °C, and an aerator was used to keep the dissolved oxygen stable and close to the saturation value. The males remained in a separate tank with aeration and temperature around 25 °C.

Reproduction was hormonally induced. Carp pituitary extract was injected in the fish, intraperitoneally, close to the pectoral fin base, using a 0.5 mL hypodermic syringe. In females, the first dose of 0.5 mg kg$^{-1}$ was applied 1 h after capture in the broodstock ponds, and the second dose of 5.0 mg kg$^{-1}$ was applied after a 12 h interval. After applying the final dose in females, a single dose of 2.0 mg kg$^{-1}$ was applied in males, and then ten males were added to each of the reproduction boxes. A single female can spawn up to 20,000 eggs. We put two males in for each female to increase the fertilization rate. The females spawned 10 h after contact with the males. Then, the eggs were collected in a 1 L beaker, and 0.1 mL samples of eggs were counted.

A total of 222,000 eggs were placed in each of two 200 L incubators. The larvae hatched within 24 h. They remained in the incubators until they absorbed the entire yolk sac and could swim in a straight line, which occurred in about 48 h. At this stage, they are called post-larvae and can be commercialized or stocked in grow-out ponds. The post-larvae were counted in one sample bowl, and the total number was estimated by comparison with

other bowls containing post-larvae. Then, the post-larvae were transferred to plastic bags and sold to grow-out farmers.

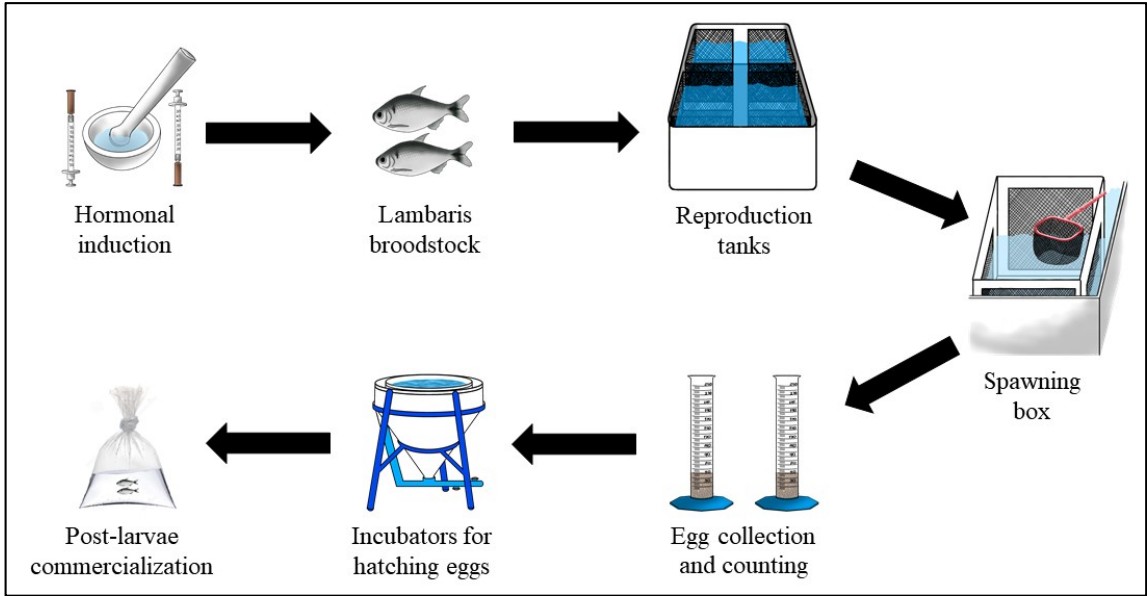

**Figure 2.** Larviculture procedures used at the Guanhanhã Municipal Aquaculture Farm.

The entire process of preparing the facilities, selecting brooders, mating, spawning, and larval development is a six-day cycle. We have produced 96,000 post-larvae of *D. iguape* in one cycle, which is consistent with the expected figures. Typically, about 100,000 post-larvae are produced in each cycle in the Guanhanhã Municipal Aquaculture Farm.

### 2.2. Economic Sustainability

The indicators of economic sustainability comprise four aspects: efficiency in the use of financial resources, the capacity of resilience, the capacity to absorb the cost of generated negative externalities, and the capacity to generate resources for reinvestment. All expenditures were estimated, including the initial investments in construction and equipment, and the operating costs per year. Gross revenues were determined considering the sales price per thousand post-larvae. We estimated 50 cycles of production per year, resulting in about 4.8 million post-larvae, according to the productivity obtained in the present study. The annual number of cycles considered one day per week to be a paid day off for the only employee.

Major positive and negative externalities were also assessed. The reduction of pollutant nutrient load (carbon, nitrogen, and phosphorus) in the inflow water, and the absorption of greenhouse gases (measured as $CO_2$ equivalents) during the *D. iguape* larviculture process were considered positive externalities. Nutrient load accumulated in the sediments and released into the water effluent, and the greenhouse gases released into the atmosphere (measured as $CO_2$ equivalents) were considered negative externalities. The costs of removing carbon, nitrogen and phosphorous from water and the $CO_2$ equivalents are quite variable worldwide. In the present work, carbon (US $30.00/t), nitrogen (US $20.00/kg), and phosphorous (US $4.00/kg) in water, were monetized according to the credit values provided by Choppin et al. [12], which have been used in aquaculture studies. The $CO_2$ equivalent values were monetized at US $25/t. This is a conservative value for 2021 from the Energy Innovation and Carbon Dividend Act, obtained at the website rff.org on 12 May 2021.

The information about investments made in the facility construction, equipment acquisition, and cost of supplies was obtained through semi-structured interviews with the facility employee and a survey of the Brazilian market. The sales price for one thousand

post-larvae of *D. iguape* considered in the present study was US $17.38, which matches the price in the park region. All values were quoted in BRL$ and converted to US$, considering the exchange rate of 18 March 2021, where BRL $5.53 equivalent to US $1.00.

### 2.3. Social Sustainability

Social sustainability was assessed considering four aspects: the principle of equity, income distribution, equal opportunities, and the generation of jobs and benefits for local communities. Indicators of racial, age and gender inclusion were not computed because the hatchery had only one employee. This employee resides at the farm with his wife. The wife does not receive remuneration from the farm, but assists in some production activities, such as counting the larvae. Semi-structured interviews were applied to the people living and working in the surrounding area and in the hatchery, to obtain data for calculating the social sustainability indicators. The indicators, measured on a scale ranged from 0 to 1, are interpreted thus: the higher the value, the greater the social sustainability.

### 2.4. Environmental Sustainability

Environmental sustainability indicators reflected four main aspects: the use of natural resources, efficiency in using those resources, the release of pollutants into the environment, and the conservation of genetic diversity and biodiversity. All calculations were carried out considering a six-day cycle of production. Samples of water, sediment, animal, and greenhouse gases were collected during the reproduction and larviculture of *D. iguape*. These samples were stored and analyzed in the Sustainable Aquaculture Laboratory—FINEP, at CAUNESP facilities at São Vicente, São Paulo, Brazil.

Three sets of water samples were collected from all the broodstock ponds, breeding, and hatchery tanks, totaling 54 sample units. Samples were obtained from the inlet, inside, and outlet water. The total carbon concentration (dissolved + particulate), dissolved inorganic carbon (sum of carbonates, bicarbonates, and carbonic gas), and total nitrogen were determined using an elemental analyzer (Elementar Vario TOC-N Select®). The total suspended solids were determined according to the method described in APHA [13] (method 2540 D). To determine the total phosphorus, the samples were previously digested, using the Persulfate Digestion Method [13] (method 4500-P B5), to release the compounds associated with organic matter in the form of orthophosphate. Then, the orthophosphate was measured by the method of stannous chloride [13] (method 4500-P D) using a digital spectrophotometer (Shimadzu UV-1800®). Six sediment samples were obtained from bloodstock ponds, breeding tanks and hatchery tanks. These were analyzed in triplicate. Sediment in the broodstock ponds was collected using Tripton samplers [14]. These are composed of six 1.876 L PVC pipes that are 9.7 cm in diameter and 25.4 cm in length. The pipes were filled with salt water, placed inside the ponds, and kept for 24 h. All sedimented material in breeding tanks was collected at the end of the breeding period by siphoning. In the hatchery, the only sedimentation observed was the shells of the hatched larvae that were siphoned. Five grams of post-larvae (~5000 fish) were randomly collected from the incubators.

To determine the carbon, nitrogen, phosphorus and energy content in sediment and fish, the samples were weighed on an analytical balance (Mettler Toledo AT21®, the accuracy of 0.1 µg), kiln-dried at 95–100 °C, and weighed again (AOAC, 1995, method 934.01). Then, carbon and nitrogen were determined using a CHNS elemental analyzer (Elementar Vario MACRO Cube®). For the phosphorus determination, the samples were previously incinerated in the muffle for 4 h at 550 °C, and the ashes were analyzed using the colorimetric method metavanadate [15]. The energy content was obtained by a calorimeter pump (IKA C2000 basic®).

Greenhouse gases were obtained and analyzed following the same procedures described in David et al. [16] and Flickinger et al. [17]. To determine the absorption and emission of $CO_2$ and $CH_4$, we collected diffusive gases in both broodstock ponds and the reproduction tanks indoors, whereas the bubbles gases were measured only in the

broodstock ponds. For the diffusive gas collection, 26 samples were obtained during the day and night to account for the effects of photosynthesis and respiration. The equilibrium method was used [18], in which an air sample collected near the surface is confined in a diffusion chamber. At the time of installation, a portion of this sample was collected and packaged in glass ampoules (with gel-sealed cap). The collection was repeated after 1, 2, and 4 min. To collect bubbles released into the atmosphere, fiberglass funnels were installed on the ponds' surface, suspended by floats. Each funnel has a 30 cm opening and an angle of 60°, ending in a diameter of 20 mm. At the end of the funnel, a connected graduated container (600 mL) collected the released bubbles over 24 h.

All gas samples were analyzed in the Permanent Gas Analyzer (Shimadzu-GC-2014®). This GC system has two sensors, the thermal conductivity detector (TCD) and flame ionization detector (FID), which quantify $CO_2$ and $CH_4$. The TCD analyzes the difference in thermal conductivity between the effluent in the column compared to the carrier gas conductivity. The FID detects organic compounds by detecting ions formed during the combustion of the compounds. The generation of these ions will be proportional to the concentration of species contained in the sample. These compounds then pass through a methane column, which transforms these compounds for FID detection. Gases are identified by the difference in column retention time, which generates a peak determined according to the column retention time.

## 3. Results

The initial investment was US $37,629 to set up a hatchery with the capacity to produce around 4.8 million post-larvae of *D. iguape* yearly. This investment can provide a return in 2.1 years, at an IRRe of almost 50%, and a profitability of US $18,310 per year (Table 1). Negative externalities totaled US $158.40, and the positive externalities, US $2.40, which resulted in an expense of US $156.00 per year. However, this expense is barely noticeable because of the high profitability. The experience of the employee in his activity is high (31 years), and the enterprise risk is low. However, the single market and single product indicate low sustainability.

**Table 1.** Indicators of economic sustainability calculated for the larviculture stage of the *D. iguape*, at the Guanhanhã Municipal Aquaculture Farm. The indicators, classified on a scale from 0 to 1, are interpreted as the higher the value, the greater the sustainability. For classified indicators by number, larger numbers indicate better results.

| Indicator | Value |
|---|---|
| Ratio between Net Income and Initial Investment (RII) (scale 0 to 1) | 0.5 |
| Internal Rate of Return (IRRe) (%) | 48.6 |
| Payback Period (PPe) (years) | 2.1 |
| Benefit-Cost Ratio (B/Ce) (US$/US $1 invested) | 1.0 |
| Net Present Value (NPVe) (US$) | 37,042 |
| Net Profit (NPe) (US$/ year) | 18,310 |
| Negative Externality (En) (US$/year) | −15,840 |
| Positive Externality (Ep) (US$/year) | 2.40 |
| Annual Income (AI) (US$/year) | 20,568 |
| Permanence of the Farmer in the Activity (PA) (years) | 31 |
| Risk Rate (RR) (scale 0 to 1) | 0.8 |
| Diversity of Products (DP) (number) | 1 |
| Diversity of Markets (DM) (number) | 1 |
| Invested Capital Generated in the Activity (ICGA) (%) | 14.7 |

The use of local workers was at maximum since the single employee lives in the region. The wage equity was 0.4 (Table 2) because the employee's wife was included in the calculations of social indicators for her unpaid assistance. However, the employee salary value was 48.4% of the total production cost. Income distribution indicates that for every US $1.00 generated for net profit, US$ 0.38 is distributed to the employee, with a

gain of US $6.03 for every thousand post-larvae produced. The systems required a low number of person-hours per area used and production unit, which was 0.9 h/m$^2$ and 0.1 h/1000 post-larvae. The larviculture process has no by-products from processing or waste. The employee does not have a private health insurance plan but accesses the public health system. The company did not provide any kind of on-the-job education, and the work safety is rated 0.6, which is slightly higher than the mean value.

**Table 2.** Indicators of social sustainability calculated for the larviculture stage of the *D. iguape*, at the Guanhanhã Municipal Aquaculture Farm. PL = post-larvae. The indicators classified on a scale from 0 to 1 are interpreted as the higher the value, the greater the sustainability. For classified indicators by number, larger numbers indicate better results.

| Indicator | Value |
|---|---|
| Development of Local Economy (LE) (scale 0 to 1) | 0.3 |
| Use of Local Workers (LW) (scale 0 to 1) | 1.0 |
| Remuneration of Work per Unit of Production (RLUP) (US$/1000 PL) | 6.03 |
| Proportion of Self-Employment (SE) (scale 0 to 1) | 0 |
| Permanence in the Activity (PA) (years) | 31 |
| Required Work per Unit of Occupied Area (WA) (person-hours/area) | 0.9 |
| Required Work per Unit of Production (WP) (person-hours/production) | 0.1 |
| Safety at Workplace (SW) (scale 0 to 1) | 0.6 |
| Local Consumption of Production (LC) (scale 0 to 1) | 1.0 |
| Pay Equity (PE) (scale 0 to 1) | 0.4 |
| Proportional Cost of Work (PCW) (%) | 48.4 |
| Income Distribution (ID) (US$) | 0.38 |
| Access to Health-Insurance Programs (AHP) (scale 0 to 1) | 1 |
| Schooling (Sc) (scale 0 to 1) | 0 |
| Participation in Outside Community Activities (PCA) (scale 0 to 1) | 0.5 |

The total area used for the larviculture stage of *D. iguape* is 405 m$^2$, corresponding to two broodstock ponds, three reproduction tanks, and two hatcheries to produce post-larvae of *D. iguape*. The crop produces around 96 thousand post-larvae per cycle (6 days), consuming 3.8 m$^3$ of water and 15.4 MJ of energy per thousand produced (Table 3). The post-larvae recover little of the energy and materials applied to the system. The pollution released in the environment was low, except for the suspended solids in the water effluents (83.3 g/1000 PL). The system can absorb greenhouse gases (18 g $CO_2$ equivalents/1000 PL) and presents no risk to the local biodiversity because the farmed species are native to the region.

**Table 3.** Environmental sustainability indicators calculated for the larviculture of the *D. iguape*, at the Guanhanhã Municipal Aquaculture Farm. PL = Post-larvae.

| Indicator | Value |
|---|---|
| **Use of Natural Resources** | |
| Use of Space (S) (m$^2$/1000 PL) | 4.2 |
| Dependence on Water (W) (m$^3$/1000 PL) | 3.8 |
| Use of Energy (E) (MJ/1000 PL) | 15.4 |
| **Materials (g/1000 PL):** | |
| Use of Carbon (C) | 16.2 |
| Use of Nitrogen (N) | 1.7 |
| Use of Phosphorus (P) | 0.02 |
| **Efficiency in the Use of Resources** | |
| Efficiency in the use of Energy (EE) (%) | 0.1 |
| Efficiency in the use of materials (%): | |
| Efficiency in the use of Carbon (EC) | 0.1 |
| Efficiency in the use of Nitrogen (EN) | 0.1 |
| Efficiency in the use of Phosphorus (EP) | 0.4 |

**Table 3.** *Cont.*

| Indicator | Value |
|---|---|
| **Release of Pollutants into the Environment (g/1000 PL)** | |
| Carbon Released in the Effluent * (CRE) | 8.5 |
| **Potential of Eutrophication (PE):** | |
| Nitrogen Released in the Effluent (NRE) | 0.5 |
| Phosphorus Released in the Effluent (PRE) | 0.10 |
| Potential of Siltation (PS) | 83.3 |
| Potential of Global Warming (PGW) (g $CO_2$ equivalents/1000 PL) | −18.0 |
| **Pollutants accumulated inside the system (g/1000 PL)** | |
| Accumulation of Carbon (AC) | 11.5 |
| Accumulation of Nitrogen (AN) | 1.0 |
| Accumulation of Phosphorus (AP) | 0.5 |
| **Conservation of Genetic Diversity and Biodiversity** | |
| Risk of Farmed Species (RFS) (scale 1 to 8; lower values are better) | 1 |

* used as a proxy of organic matter.

## 4. Discussion

The *D. iguape* larviculture showed potential for being set up near conservation areas, to promote the local communities' sustainable development through aquaculture. The case analyzed in the present study indicates attractive economic gains and the rapid return of the invested capital. The initial investment is about US $40,000, which is high for rural farmers, but low for public or private investors. The hatchery studied generated few direct jobs, but the workforce can be recruited from the community. A hatchery can enable the establishment of several small lambari grow-out farms, leveraging the development of indirect jobs through self-employment. The production system studied had a low environmental impact, including the minor release of nutrients in the water effluent and accumulation on the pond bottom. Strategies for using the effluent and the sediment as fertilizer can be developed as mitigating actions, increasing the company's re-use of waste and, consequently, the circularity [11]. In addition, the *D. iguape* hatchery studied absorbed greenhouse gases and produced a species native to the Sea Mountains State Park. Therefore, a *D. iguape* hatchery has the potential of combining biodiversity conservation and income generation, meeting the Sustainable Development Goals (SDGs) numbers 1, 6, 13, 14, as defined in the Agenda 2030, that are No Poverty, Clean Water and Sanitation, Climate Action, and Life Below Water, respectively [19].

The Peruibe City Hall owns the hatchery described in this study. The municipality government invested funds for building the facilities and purchasing the equipment as the initial investment. The City Hall also funded the operational costs. However, the invested capital can give a return rapidly, and the revenues cover production expenses. Thus, this hatchery maintains itself, and data indicate it is lucrative without public subvention. The larviculture studied can provide an annual net profit of US $18,310, which is 7.7 times more than the annual Brazilian minimum wage (US $2387) and even 1.2 times higher than the North American minimum wage (US $15,080). Frequently, hatcheries need subsidies from governments to promote aquaculture development [20]. Therefore, results obtained in the present study suggest that the studied hatchery of *D. iguape* has an economic advantage compared to others around the world.

The studied hatchery of *D. iguape* may be characterized as a business model with an initial investment of about US $40,000, which returned in ~2 years, even including the negative externality costs. The business presumably also generates funds for future reinvestment in expansion, upgrades, or technology modernization. This scenario suggests an environmentally friendly business that promotes conservation, may alleviate poverty, and improve rural communities' quality of life in the light of the SDGs [19]. Nevertheless, such a result should be carefully analyzed because it comes from a descriptive study of a single hatchery, presuming constant productivity. In addition, complementary studies



should be done to demonstrate if the grow-out phase of *D. iguape* is economically feasible and can be performed with low environmental impact.

The inclusion of the externalities is scarce in aquaculture economic analyses. An example is the sustainability assessment of the macroalgae *Hypnea pseudomusciformis* culture [21]. Nevertheless, measuring externalities is essential to demonstrate if the production system remains profitable after paying for the damages caused to the environment or to third parties. In addition, externalities provide essential information for public policies related to taxes or compensations [4]. In the present study, the hatchery of *D. iguape* showed negative externalities of about US $150 yearly, which is much lower than the net profit of about US $18,000. This indicates that the adverse effects generated by production can be offset without significantly altering the annual net profit.

The economic feasibility of a hatchery depends on developing the grow-out production chain of the species produced [22]. Hatcheries are business-to-business companies. Therefore, mapping the potential clients and establishing associated policing to develop the grow-out farms is essential. In addition, it is necessary to develop an effective supply chain and expand the market for trading adult fish in the region to assure the success of grow-out farmers. This process needs effective local governance. The market for adult *D. iguape* is diverse, which increases the opportunities for rural grow-out farms. The lambari can be sold as appetizers for consumption in homes, bars, restaurants, beach kiosks, as nutritional food for humans, live food for carnivorous ornamental species, and live bait for sport-fishing. *Deuterodon iguape* is saline-tolerant and can replace shrimp-baits in marine sport-fishing [23]. This species can also be used in commercial tuna fisheries as a bait alternative, minimizing the pressure on natural stocks of sardines for this purpose [22]. Studies carried out in Brazil showed great acceptance of lambari as a nutritional food source [5]. Fish consumption offers a better protein source than birds, cattle, and swine [24]. Small farmers could set up cooperatives and obtain funds to improve the processing and packaging of the lambari, to reach more distant markets.

The indicators of social sustainability detected strengths and weaknesses in the production system. The strengths include hiring the workforce from the local community and the development of the local economy. The tasks to produce *D. iguape* post-larvae are simple and can be performed by persons from the local community after training. About 30% of the larviculture supplies were purchased locally, and 100% of the post-larvae produced supplied the local grow-out farms within or surrounding the Sea Mountains State Park. Thus, the lambari hatchery positively impacts the regional economy and consequently allows socio-economic development for the local populations. Access to local markets and appropriate tools are necessary for aquaculture to generate income and employment for rural communities [25,26]. This strategy should be linked to small grow-out producers that buy the post-larvae. The farms located within conservation areas face a lack of basic infrastructures, such as road maintenance, which hampers product distribution. Therefore, the development of lambari hatcheries should be part of public policies to promote sustainable development based on the local necessities. The weaknesses detected by the social indicators were low education, the lack of private health plans, and regular work security. The workplace needs improvement, since education and health are the constitutional rights of Brazilian citizens.

The employee's annual income of US $6949 is sufficient to support his family. The employee's salary represents half of the total production cost, and is 3.4 times higher than 57.6% of Brazilians, who live on US $2055 as annual income [27]. However, the salary equity indicator was low because the employee's wife does not receive a salary but assists him in the larviculture activities. This situation is common in several rural communities, in which family members work without payment [28]. This is an important social issue that must be solved, and all employees should be adequately remunerated, regardless of gender or kinship, according to SDG 5, Gender Equality.

The indicators of environmental sustainability showed some weaknesses in the production of post-larvae of *D. iguape* at the Guanhanhã Municipal Aquaculture Farm. The

system has low efficiency in the use of energy (0.1%) and materials (N and C = 0.1% and $p$ = 0.4%). These results may be related to the low mass of the post-larvae (0.001 g). Therefore, the inputs are disproportionate to the outputs, and the system should be modified to improve efficiency in using natural resources. In addition, for every thousand post-larvae produced (1 g of product), 8.5 g of carbon, 0.5 g of nitrogen, 0.1 g of phosphorus, and 83.3 g of total dissolved solids are released into the environment by water effluents; similarly, 11.5 g carbon, 1 g nitrogen, and 0.5 g phosphorus were accumulated in the pond bottom. Small adaptations to the facilities and management can partially solve these weaknesses. The introduction of circular economy concepts in the hatchery system can promote the use of effluents and sediments as fertilizers for agriculture or even for aquaculture [11]. Water quality and flow are crucial factors that should be controlled for aquaculture systems' success [29,30].

The low environmental risk of cultivated *D. iguape* and the absorption of greenhouse gases were the central environmental hatchery's strengths. The culture of a native species combines economic activity with biodiversity conservation. The culture of *D. iguape* in the surrounding areas of the Sea Mountains State Park was introduced to replace the culture of the Nile tilapia (*Oreochromis niloticus*). This exotic species is the principal fish farmed in Brazil [22], but the environmental risk is level 5 [31] on a scale from 1 to 8 [4], while *D. iguape* is level 1. Therefore, the larviculture of this lambari is essential to supply post-larvae and maintain the farming of a native species in the areas surrounding the park. In addition, the larviculture of *D. iguape* absorbed 18 g of $CO_2$ equivalent per thousand post-larvae produced. The hatchery studied operates in an open system, which requires a constant flow of water to keep high oxygen saturation. The water pumped from a river contains plankton. Therefore, it is likely that $CO_2$ is absorbed by the photosynthetic process inside the ponds and tanks, and the rate of assimilation surpasses the rate of decomposition of organic matter. If confirmed in more complete investigations, the positive externalities, such as the absorption of greenhouse gases, can be considered in rewarding payment programs for environmental services.

The present study analyzed a single hatchery (case study). This is certainly a limitation and, thus, the results, inferences and generalizations should be carefully analyzed. Nevertheless, the data are consistent, and as far as we know, this is the first study using sustainability indicators to characterize a hatchery of an aquatic species. Therefore, these first data are an important start to sustainability studies, and may support further research.

In conclusion, the present study described major features of the economic, social, and environmental characteristics of a *D. iguape* hatchery, which is a critical element of the supply chain. The results also support the hypothesis that the installation of native-fish hatcheries close to protected areas may be a profitable and sustainable action to leveraging aquaculture as an economic activity. Further studies should be developed to confirm the present conclusions. Besides, complementary investigations addressing the governance and the institutional dimensions are essential to understand other important aspects, such as the dependency of public infrastructure and technical assistance services to the structure of the production chain.

**Author Contributions:** Conceptualization, D.B. and W.C.V.; methodology, D.B., F.H.G. and W.C.V.; software, D.B.; validation, D.B., F.S.D., F.H.G. and W.C.V.; formal analysis, D.B., F.H.G. and F.S.D.; investigation, D.B.; resources, W.C.V.; data curation, D.B.; writing—original draft preparation, D.B.; writing—review and editing, F.H.G. and W.C.V.; visualization, D.B., F.S.D., F.H.G. and W.C.V.; supervision, W.C.V.; project administration, D.B. and W.C.V.; funding acquisition, W.C.V. All authors have read and agreed to the published version of the manuscript.

**Funding:** This research was funded by São Paulo Research Foundation—FAPESP, processes # 10/52210-3 and 2015/19451-8 and by National Council for Scientific and Technological Development— CNPq, processes # 562820/2010-8, 406069/2012-3, and 306361/2014-0; CAPES-EMBRAPA public notice: 15/2014, project number 24; FINEP, agreement n° 01.10.0578.00/10.

**Institutional Review Board Statement:** The study was conducted according to the guidelines of the Declaration of Helsinki, although it was not submitted to an Institutional Review Board.

**Informed Consent Statement:** Informed consent was obtained from all subjects involved in the study.

**Acknowledgments:** The authors thank the Agricultural Department of the City of Peruibe, especially Darci Spinoza, the only employee of the facility, for collecting the data used to calculate the sustainability indicators.

**Conflicts of Interest:** The authors declare no conflict of interest.

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
