# Peer review of "Sustainability Analysis of the Production of Early Stages of the Atlantic Forest Lambari (Deuterodon iguape) in a Public Hatchery at a Rainforest Conservation Area"

_sustainability, doi:10.3390/su13115934_

Round 1
Reviewer 1 Report
An interesting and well-written study that gives a good overview of the possibilities for sustainable use of aquaculture even in protected areas.
Nevertheless, there is one point that as a reviewer I found very surprising and thought-provoking, and that is the very low number of replicates in relation to the reproductive events.
From the authors' point of view, how was it ensured that the present result was representative and reproduceable? This is all the more astonishing to me because, according to the description in the manuscript, the complete reproduction cycle takes only one week.
Thus, a single or multiple repetition would not have been a great challenge in terms of time.
The authors should include a detailed explanation here, as the rest of the manuscript clearly assumes that the results can be repeated like this up to 50 times a year.
Otherwise, the number in line 247 for the unit m² should be raised.
In the bibliography, the species names are to be written kurisv for sources no. 12., 17. & 30.
Author Response
Open Review 1
Comments and Suggestions for Authors
-An interesting and well-written study that gives a good overview of the possibilities for sustainable use of aquaculture even in protected areas.
Author's answer Thanks for your incentive.
-Nevertheless, there is one point that as a reviewer I found very surprising and thought-provoking, and that is the very low number of replicates in relation to the reproductive events. From the authors' point of view, how was it ensured that the present result was representative and reproduceable? This is all the more astonishing to me because, according to the description in the manuscript, the complete reproduction cycle takes only one week. Thus, a single or multiple repetition would not have been a great challenge in terms of time. The authors should include a detailed explanation here, as the rest of the manuscript clearly assumes that the results can be repeated like this up to 50 times a year.
The author's answer: That is a good point. In fact, we have observed the reproduction and larviculture of the D. iguape a few times at the Guanhanhã Municipal Aquaculture Farm and realize that it is a simple and repeatable process with consistent results in the different cycles. These observations were confirmed by the operator, who has been working there for many years. The cycle we have used to obtain data showed productivity similar to that usually obtained. In addition, our results are corroborated by former reproductive studies for this same species published in Portuguese in local papers, which we decided do not cite in the MS to avoid use gray literature. Thus, we assumed that this cycle was representative of the process. It should be noted that the cycle is short (6 days), but the sample analyses are complex and very expensive. We agree with the referee that the analyses of more than one production cycle could add more credibility to the results. Nevertheless, we are confident that our data are sound and represent the usual scenario. We have added an explanation on the MS to support the use of only one cycle.
-Otherwise, the number in line 247 for the unit m² should be raised.
The author's answer: It was revised.
-In the bibliography, the species names are to be written kurisv for sources no. 12., 17. & 30.
Author answer: They were revised.
Reviewer 2 Report
Dear authors:
Review Comments on Manuscript entitled " Sustainability analysis of the production of early stages of the Atlantic forest lambari (Deuterodon iguape) in a public hatchery at a rainforest conservation area" (sustainability-1188867 ):
Please consider the following points during the revision of the manuscript:
The present work is of interest to a broad audience since it studies several environmental, socio-economics factors that may lead to sustainable aquaculture farms in close balance with nearby protected areas in rainforest areas.
MM
Although the authors refer to the article where they based on to select the indicators and calculations, it will be helpful to provide a full description in Supplementary Material with the actual values for this work.
Author Response
Open Review 2
Comments and Suggestions for Authors
-The present work is of interest to a broad audience since it studies several environmental, socio-economics factors that may lead to sustainable aquaculture farms in close balance with nearby protected areas in rainforest areas.
Author's answer: Thanks for your encouraging comment.
-Although the authors refer to the article where they based on to select the indicators and calculations, it will be helpful to provide a full description in Supplementary Material with the actual values for this work.
The author's answer: The full description is presented in the Valenti et al. (2018), which is an open-access article, and thus, easily available. Thus, we think that it is not necessary to repeat it using Supplementary Material.
Reviewer 3 Report
The topic of this manuscript is interesting. However, the study is about a single farm and a case study is not a sustainability study. There is not a descriptive statistics.
The abstract is very basic and lacks of data.
I miss many references along the manuscript. For example in lines 55-58, 75-78, 110-135.
Lines 100-105 repeat information included in subsections 2.2, 2.3 and 2.4.
Subsection 2.1 should be divided in paragraphs. I miss a full stop in lines 113, 115, 118...
Avoid to use the first person (we).
Delete lines 139-141. This subsection is not an introduction.
The reference [12] for monetized carbon is 11 years old.
investing.com is not an aproppiate reference (line 155).
The numbers of samples are missed in lines 176-177.
What is the sampling method for GHG in line 177?
Results and discussion are not solid with just a single case. For example lines 241-251 look a farmer story more than a research.
I suggest to study more farms to have descriptive statistics or use data about release of pollutants into the environment for a short communication in other journal.
There is a lack of conclusions.
Author Response
Open Review 3
Comments and Suggestions for Authors
-The topic of this manuscript is interesting. However, the study is about a single farm and a case study is not a sustainability study. There is not a descriptive statistics.
Author's response: We respectfully disagree. The indicators of sustainability can be used on farm, region, country or global levels. The application to access one farm is a case study and it may be very useful in the onset research fields like the study of aquaculture sustainability. In addition, statistics are not commonly used in sustainability analyses.
-The abstract is very basic and lacks of data.
Author's response: Please, look at the abstract again and you will see that the abstract shows many data.
-I miss many references along the manuscript. For example in lines 55-58, 75-78, 110-135.
Author's answer: We respectfully disagree. We can support these claims using our own observations. Lines 110-135, for instance, contain the description of our procedures and thus do not need references. Some information not referenced can be found in local publications in Portuguese, which does not have easy access to the readers. We avoided using these grey references.
-Lines 100-105 repeat information included in subsections 2.2, 2.3 and 2.4.
Author's answer: Thanks for your observation. We have deleted it.
-Subsection 2.1 should be divided in paragraphs. I miss a full stop in lines 113, 115, 118...
Author's answer: Thanks for your observation. We totally agree and have split it into four paragraphs.
-Avoid to use the first person (we).
Author's answer: This a question of style and we supposed that the use of the first person is accepted in the Sustainability journal. Anyway, we have changed when it was possible.
-Delete lines 139-141. This subsection is not an introduction.
Author's answer: We decided to retain the lines 139-141 because such information is not an introduction but describes what the indicators used in the present study measure. Thus, we think that it is an important part of the method. We have changed the text in the MS to make it clearer.
-The reference [12] for monetized carbon is 11 years old.
Author's answer: We agree; thanks for calling our attention to this and allow us to add an explanation on the MS. The monetary values for remoting carbon, nitrogen and phosphorus from the water, as well as to the value of CO2 equivalent are quite variable in the world (See World Bank 2019 for details). Thus, we decided to use the values proposed by Chopin et al. (2010) that have been used in aquaculture (see Pereira et al., 2021 for example) to permit comparisons with other studies.
(World Bank Group, 2019. State and Trends of Carbon Pricing 2019, State and Trends of Carbon Pricing 2019. Washington, DC. https://doi.org/10.1596/978-1-4648-1435-8.)
-investing.com is not an appropriate reference (line 155).
Author's answer: We agree and have modified the MS.
-The numbers of samples are missed in lines 176-177.
Author's answer: Thanks for this observation. We have included the number of samples in the description of the sampling method of each variable.
-What is the sampling method for GHG in line 177?
Author's answer: Please, observe that this method is well described and referenced in lines 206 to 227.
-I suggest to study more farms to have descriptive statistics or use data about release of pollutants into the environment for a short communication in other journal.
Author's answer: Study more farms is impossible because this one is the unique fish hatchery located in Conservation areas in Brazil. In addition, we think that the data obtained is sound and new. As long as we know, our study is the first one focus on a hatchery. Therefore, it is really important, has novelties and new insights. Thus, we are sure that it matches the standard to be published in the Sustainability journal.
-There is a lack of conclusions.
Author's answer: We change the last paragraph to highlight the conclusion.
Round 2
Reviewer 3 Report
Although the authors do not agree with the reviewer, this study is about a single farm and a case study is not a sustainability study. There are not descriptive statistics.
There are more comments that have not been accepted by the authors but I am not going to discuss them
This manuscript lacks the quality to be published in this journal.
It is not worth adding more comments because it is not possible to improve it.